# Vitamin A Nutritional Status Is a Key Determinant of Bone Mass in Children

**DOI:** 10.3390/nu14214694

**Published:** 2022-11-06

**Authors:** Xuanrui Zhang, Jiapeng Huang, Yingyu Zhou, Zhen Hong, Xiaoping Lin, Shanshan Chen, Yongnong Ye, Zheqing Zhang

**Affiliations:** 1Department of Nutrition and Food Hygiene, Guangdong Provincial Key Laboratory of Tropical Disease Research, School of Public Health, Southern Medical University, Guangzhou 510515, China; 2Pharmacy Department, Guangzhou Panyu Hospital of Traditional Chinese Medicine, Guangzhou 510000, China

**Keywords:** retinol, bone, children

## Abstract

The bone mass increases that occur during the period of childhood are of great significance for maximizing the peak bone mass in adults and preventing for osteoporosis. Studies have reported that VA can improve the bone health in adults. Moreover, limited studies have assessed such associations in children. In this cross-sectional study including 426 children, we assessed the children’s plasma retinol concentration by liquid chromatography–mass spectrometry and the dietary intake of VA and carotenoids using a structured Food Frequency Questionnaire. Their bone mineral content and bone mineral density (BMD) were measured using dual-energy X-ray absorptiometry. After adjusting for potential confounders, the restricted cubic spline revealed an inverted U-shaped association between plasma retinol concentration and BMD; the estimated effects on the TBLH BMD per μmol/L increase in the plasma retinol concentration were 1.79 × 10^−2^ g/cm^2^ below 1.24 μmol/L and −5.78 × 10^−3^ g/cm^2^ above this point (*p* for non-linearity = 0.046). A multiple linear regression analysis revealed a positive association between the plasma retinol concentration and the TBLH BMC (*β* = 1.89, 95% CI: 1.64 × 10^−1^–3.62, *p* = 0.032). In conclusion, an appropriate plasma retinol concentration and greater intakes of dietary VA and β-carotene may enhance the bone mineral status of children who are aged 6–9 years.

## 1. Introduction

Osteoporosis is a systemic bone disease that develops when the bone mineral density and quality decrease, the bone microstructure is destroyed, and bone fragility increases [1]. In recent years, osteoporosis has gradually become a serious public health concern [2]. In China, the number of major osteoporotic fractures (wrist, vertebral body, and hip) was approximately 2.69 million in 2015, and this number is estimated to increase to 4.83 million in 2035 and 5.99 million in 2050 [3]. The bone mass increasing and the bone tissue accumulating during the period of childhood are of great significance for maximizing the peak bone mass in adults and for preventing osteoporosis [4]. Therefore, it is extremely important to maintain and maximize bone health during childhood in order to reduce the risk of osteoporosis-related diseases and fractures later in adulthood.

Studies have reported that bone homeostasis is influenced by many dietary factors, including vitamin A (VA) [5,6,7,8,9]. VA, which is mainly derived from animal sources such as liver, dairy, and egg yolks or plant-based sources enriched with carotenoids such as carrot, mango, and pumpkin (e.g., α-carotene, β-carotene, and β-cryptoxanthin), is a fat-soluble antioxidant and an essential nutrient which is of great importance for improving children’s innate and adaptive immune responses (leading to a lower susceptibility to infections, the reduction of complications, and a better prognosis), maintaining normal vision, the enhancement of growth, and cell differentiation and reproduction [10,11,12,13,14,15]. Moreover, VA can affect different stages of osteogenesis by enhancing early osteoblast differentiation and inhibiting bone mineralization via retinoic acid receptor signaling and by regulating the osteoblast- and osteoclast-related peptides [9]. The antioxidant properties of VA help to protect the bone cells against the effects of free radicals, which can otherwise cause bone resorption and inhibit osteogenic differentiation [14]. However, population-based studies have demonstrated heterogeneity in the associations of blood retinol concentration and dietary VA intake with bone mineral status. For example, some studies have shown that the circulating concentration of retinol or the dietary intake of VA and carotenoids are positively associated with bone health [16,17,18,19,20,21,22], whereas others have detected U-shaped or even negative associations [20,23,24]. Moreover, only limited studies have assessed the above associations in pediatric populations.

Therefore, in this study, we aimed to assess whether the plasma retinol concentration, dietary VA intake, and carotenoids intake are positively associated with the bone mineral status among healthy children who were aged 6–9 years.

## 2. Materials and Methods

### 2.1. Subjects

This cross-sectional study was approved by the ethics committee of the School of Public Health at Sun Yat-sen University (No. 201549). Written consent was obtained before the enrollment for each participant through his or her parent or legal guardian. We recruited healthy children who were aged 6–9 years from kindergartens and primary schools in Guangzhou between 2015 and 2017 [25]. The recruitment strategy involved distributing leaflets to primary schools in five areas of Guangzhou and contacting the mutual acquaintances of parents through public WeChat accounts. In total, 1600 healthy children were invited to participate in the study, 493 of whom responded and agreed to participate in the study. Based on the following criteria, 67 children were further excluded: (a) twins and those who underwent a preterm birth; (b) those with incomplete general data or incomplete information regarding bone mineral content and density or plasma retinol concentration; (c) those with a history of serious disease or disability. Finally, a total of 426 singleton-birth children (183 girls and 243 boys) who were aged 6–9 years were included in this study.

### 2.2. Measurement of Bone Mineral Content, Bone Mineral Density, and Body Fat Percentage

Whole-body dual-energy X-ray absorptiometry (DXA) was performed using the Hologic Discovery W System (Discovery W; Hologic Inc., Waltham, MA, USA) according to the manufacturer’s instructions to determine the total body (TB) and the total body less head (TBLH) bone mineral content (BMC), the bone mineral density (BMD) as well as the total body fat (TBF). The coefficients of variation between five consecutive measurements of the TB BMC (BMD), TBLH BMC (BMD) and TBF conducted on 33 children on the same day were 1.09% (1.58%), 1.37% (2.04%) and 1.7%, respectively.

### 2.3. Measurement of Plasma Retinol Concentration and Dietary Vitamin A Nutritional Status

Venous blood samples were obtained from the children in the morning after a 10 h of fasting. The plasma was separated within 2–4 h and stored at −80 °C until the analysis was performed. The plasma retinol was isolated and quantified using liquid chromatography–mass spectrometry which was conducted using the Qlife Lab 9000plus system (Qlife Co., Ltd., Nanjing, China) with the MassHunter Workstation Data Acquisition platform (Agilent Inc., Palo Alto, CA, USA). A pooled plasma sample was analyzed with batches of study samples to monitor them with analytical precision, with approximately day-to-day coefficient of variation values of 7.7%.

The children’s dietary intake over the past year was assessed using a structured Food Frequency Questionnaire (FFQ) involving 79 food items that were classified into 20 food groups. The dietary intakes of energy, protein, calcium, phosphorus, VA, and β-carotene were calculated using the 2009 China Food Composition Table [26].

### 2.4. Covariates

Detailed sociodemographic data (delivery mode, feeding patterns during infancy, household income, and parental educational level) and information regarding dietary habits (calcium and multivitamin supplementation) were obtained through the face-to-face interviews of parents and children, and these were conducted by trained interviewers. The child delivery mode was defined as a binary variable: cesarean or vaginal. Their feeding patterns during infancy were classified as breastfeeding or artificial feeding. The household income per month was classified into four categories: ≤8000 Yuan, 8000–15,000 Yuan, >15,000 Yuan, and unknown. The parental education level was classified as secondary or less, university, and postgraduate or above. The weights and heights were measured while the children were wearing light clothes and no shoes. Their physical activity was assessed using a 3-day physical activity questionnaire and calculated by combining the metabolic equivalent (MET, kcal·kg^−1^·h^−1^) for each type of physical activity after multiplying it by its duration per day [27].

### 2.5. Statistical Analysis

The continuous variables are expressed as means ± standard deviations (SDs) if they are normally distributed or as medians with interquartile ranges if they are not normally distributed. The categorical variables are presented as percentages. The continuous variables were assessed using Student’s *t*-test, and the categorical variables were assessed using the χ^2^ test to assess the differences in the characteristics between the girls and boys.

The dietary protein, calcium, phosphorus, VA and carotenoids intake levels were adjusted using the residual energy adjustment method. The regression of restricted cubic spline with three knots at the plasma retinol of 10th, 50th and 90th centiles was used to flexibly model the association between the plasma retinol concentration and the bone mineral status after adjusting for age, gender, height, TBF, delivery mode, feeding pattern, maternal and paternal educational level, household monthly income, use of calcium and multivitamin supplement, physical activity, total energy, and energy adjusted dietary intakes of protein, calcium and phosphorus. Multiple linear regression models using the enter method were applied to examine whether the dietary VA (carotenoids) intake were significantly associated with the children’s BMC and BMD. Model 1 was only adjusted for age and sex; model 2 was adjusted using the all of the same confounders as were used in the restricted cubic spline. 

All of the statistical analyses and plots were performed using R version 4.1.2. The significance level was set at 0.05.

## 3. Results

### 3.1. Characteristics of the Children

In total, 426 children who were aged 8.0 ± 1.0 years were included in this cross-sectional study. As presented in Table 1, the means ± SDs of the TB BMC, TBLH BMC, TB BMD and TBLH BMD were 930 ± 138 g, 585 ± 111 g, 0.781 ± 0.063 g/cm^2^ and 0.609 ± 0.063 g/cm^2^, respectively. The plasma retinol concentration, dietary VA, α-carotene, β-carotene, and β-cryptoxanthin daily intakes were 1.11 ± 0.28 μmol/L, 676 ± 354 μg RE, 860 ± 883 μg, 2408 ± 1615 μg and 97.6 ± 103 μg, respectively.

### 3.2. Regression of Restricted Cubic Spline of the Association between Plasma Retinol Concentration and BMD and BMC 

In Figure 1, we used restricted cubic splines to flexibly model and visualize the relationship between the predicted plasma retinol concentration and the bone mineral indexes. The effect of plasma retinol on the TBLH BMD was positively steep, rising (estimated average effect = 1.79 × 10^−2^ g/cm^2^, 95% CI: 4.21 × 10^−3^–3.16 × 10^−2^ g/cm^2^) until 1.24 μmol/L, and then, it started to decline (estimated average effect = −5.78 × 10^−3^ g/cm^2^, 95% CI: −2.08 × 10^−2^–9.26 × 10^−3^ g/cm^2^) afterwards (*p* for non-linearity = 0.046, *p* for overall = 0.037). Consistently, there was also an inverted U-shaped relationship which was observed for the TB BMD (*p* for non-linearity = 0.001, *p* for overall = 0.004); the estimated average effects per μmol/L increase in the plasma retinol concentration increase were 3.05 × 10^−2^ g/cm^2^ (95% CI: 1.15 × 10^−2^–4.95 × 10^−2^ g/cm^2^) below 1.17 μmol/L and −2.37 × 10^−2^ g/cm^2^ (95% CI: −4.58 × 10^−2^ – −1.52 × 10^−3^ g/cm^2^). These similar association did not have statistical significance for the TB BMC (*p* for non-linearity = 0.053, *p* for overall = 0.056). However, a significantly positive linear relationship between the plasma retinol concentration and the TBLH BMC was detected; per μmol/L increase in the plasma retinol concentration resulted in a 1.89 g (95% CI: 1.64 × 10^−1^–3.62 g) increase in the TBLH BMC (*p* = 0.032).

### 3.3. Multiple Linear Regression Analysis of the Association between VA and BMD and BMC

As presented in Table 2, after the adjustment for all of the confounders in model 2, an increase of 1 SD in the dietary intake of VA resulted in a 5.86 × 10^−3^ g/cm^2^ (95% CI: 1.56 × 10^−3^–1.02 × 10^−2^ g/cm^2^) increase in the TB BMD (*p* = 0.008). The TB BMD increased by 4.47 × 10^−3^ g/cm^2^ (95% CI: 5.47 × 10^−5^–8.88 × 10^−3^ g/cm^2^) with every 1 SD increase in the dietary β-carotene intake (*p* = 0.048). However, no significant association was observed between the dietary α-carotene, β-cryptoxanthin intake and the bone mineral status.

## 4. Discussion

This study evaluated the associations of plasma retinol concentration and dietary VA and carotenoids intake with the bone mineral status in children who were aged 6–9 years. Additionally, the results indicated that vitamin A nutritional status was a key determinant of bone mass in children.

### 4.1. Associations of Plasma Retinol Concentration with Bone Mineral Status

Maggio et al. assessed 90 participants who were aged ≥ 55 years and reported that the plasma retinol concentration was lower among women with osteoporosis (2.020 μmol/L) than it was among the controls (2.290 μmol/L) (*p* < 0.001) [19]. Another recent study that excluded Hispanic populations reported that every unit increase in the serum retinol concentration resulted in a 0.011 g/cm^2^ increase in the BMD of both the femur neck and the entire hip [18]. In contrast, some studies have indicated that a high serum retinol concentration could have detrimental effects on the bone mineral status. A cross-sectional study including 229 Spanish postmenopausal women with an average age of 57.4 years old reported that osteoporotic women who had the highest serum retinol concentration (3.35–5.64 μmol/L) had an up to 8× (odds ratio [OR] = 8.37, 95% confidence interval [CI] = 2.51–27.91) greater risk of developing osteoporosis than those who had the lowest serum retinol concentration (1.17–1.98 μmol/L) [23]. Michaëlsson et al. reported that the risk of a fracture in the highest quintile of the serum retinol concentration (>2.64 μmol/L) was 1.64 times (95% CI = 1.12–2.41) higher than that in the middle quintile (2.17–2.36 μmol/L) [24]. Our results indicate an invert U-shaped association between the plasma retinol concentration and BMD among children, whereas a positive linear relationship was observed between the plasma retinol concentration and the TBLH BMC. The inflection point of plasma retinol concentration on bone health (TBLH BMD: 1.24 μmol/L; TB BMD: 1.17 μmol/L) was approximately 1.2 μmol/L, which was lower than that which has been reported in studies stating that there are negative associations between the plasma retinol concentration and the bone mineral status [23,24]. In addition to the difference in the blood level of plasma retinol, other factors, including the participant age, study design, bone health measures, and the patient’s genetic backgrounds may have collectively contributed to the heterogeneity of the findings. 

### 4.2. Dietary VA Intake

In terms of the dietary VA intake, which is consistent with our study, a case–control study including 1452 elderly Chinese patients with newly diagnosed hip fractures and control participants reported that a high dietary VA intake (females: 673 μg/day; males: 678 μg/day) was associated with a reduced risk of hip fracture (OR = 0.63, 95% CI = 0.42–0.98) [22]. However, a cohort study including 570 women and 388 men aged 55–92 years at baseline reported an inverted U-shaped association between the dietary VA intake and the BMD. A dietary VA intake of 0–600 μg/day (2000 IU/day) was positively associated with the BMD; in contrast, an inversed association was noted if the intake was beyond 840 μg/day (2800 IU/day) [20]. Moreover, we observed a favorable association between the β-carotene intake and the children’s BMD. In a case–control study involving 1070 patients who were aged 55–80 years with hip fractures and 1070 sex-matched controls, the OR of a fracture in the highest quartile of α-carotene intake was 0.45 (95% CI = 0.30–0.66) in comparison with that of the lowest quartile [16]. Consistently, a cross-sectional study including 8022 participants who were aged 30–75 years reported that postmenopausal women in the highest quintile (7.20–87.12 mg/day) of β-carotene intake had a 63% (OR = 0.37, 95% CI = 0.15–0.93) lower risk of osteopenia than those in the lowest quintile (0.0009–0.70 mg/day). In addition, every unit increase in the dietary β-cryptoxanthin intake resulted in a 0.003 g/cm^2^ increase in the total hip BMD of the premenopausal women (*p* = 0.026) [21]. However, in our study, we did not observe any such effect of α-carotene or β-cryptoxanthin intake on the bone mineral status, possibly because of the small sample size or because the dietary β-cryptoxanthin intake was low among our study subjects.

### 4.3. Mechanisms

The following mechanisms may explain the associations that we observed in our study. An adequate VA intake (retinol or retinoic acid) is of great importance in maintaining bone homeostasis, which effectively suppresses the over-secretion of the parathyroid hormone [28,29]. It may also induce the secretion of growth hormones to stimulate the production of insulin-like growth factor 1; both of these compounds promote osteoblast proliferation and differentiation and reduce osteoblast apoptosis by stabilizing β-catenin in the Wnt signaling pathway [30,31,32]. However, VA is a fat-soluble vitamin, and its long-term excessive intake will accumulate in the body and cause toxic side effects such as alopecia, cheilosis, liver injury and so on [33,34]. Previous studies in vivo and in vitro have indicated that a high concentration of preformed VA negatively affects both osteoblast differentiation and mineralization, owing to the effects of VA on osteogenic gene inhibition, osteoclastogenic gene activation, and osteocyte or osteoblast-related peptides modulation [9]. Through an animal experiment using female C57BL6/J mouse models, researchers reported a significant increase in the osteoclast number and the osteoclastogenic gene expression in mice that were treated with high doses of dietary retinyl acetate [35]. Moreover, VA antagonizes vitamin D (VD) by increasing calcium absorption and maintaining a steady-state serum calcium concentration. Retinoic acid and 1, -25-hydroxyvitamin D share a common nuclear receptor that interacts with both the retinoic acid receptor and the VD receptor. Therefore, a high plasma retinol concentration may reduce the function of VD and negatively affect bone homeostasis [36]. These facts may provide potential explanations for the inversed U relationship between the plasma retinol concentration and the bone mineral status in our study. The majority of the previous studies suggested that oxidative stress which is caused by reactive oxygen species can promote bone resorption and inhibit osteogenic differentiation [14]. These processes can negatively affect bone homeostasis, reduce the bone mass, and even lead to osteoporosis [37]. Carotenoids act as antioxidants that quench singlet oxygen and trap peroxyl radicals, reducing the oxidative stress and potentially reducing the bone resorption, thereby protecting the bone tissue [14,38]. However, the oxidation capacities of various carotenoids are different [39]. β-carotene is one of the most abundant carotenoids in the diet which has the highest number of biological activities for the conversion to its retinol equivalents [40]. It can also inhibit osteoclastogenesis and bone resorption by suppressing the receptor activator of the nuclear factor kappa-B ligand signaling pathway [41]. These mechanisms may collectively contribute to the protective effects of dietary carotenoids intake (especially for β-carotene) on bone health.

### 4.4. Strength and Limitations

To the best of our knowledge, this study is the first to comprehensively describe the associations of plasma retinol concentration and dietary VA (carotenoids) intake with bone mineral status among children who are aged 6–9 years. 

However, there exist several limitations to our study. Firstly, the causality of the association between VA and bone mineral status could not be accurately demonstrated using the cross-sectional study design. Secondly, some nutrients from the supplements or fortified foods were not included in the calculation of the dietary nutrient intake in our analyses. However, the use of multivitamins and calcium supplements were applied as binary categorical variables for adjustment. Thirdly, the carotenoids contents in foods may differ, depending on the plant’s cultivation, harvesting, processing and storage conditions. Moreover, the dietary VA intake was calculated using the 2009 China Food Composition Table, and thus, the findings may not be generalizable to other countries. This may also have contributed to the heterogeneity among the different studies. Fourthly, our sample size was small, and all of the participants were from Guangzhou and they had a narrow age range, thus, caution should be exercised when one is extrapolating our results to children in other areas and of different ages. Fifthly, although many covariates were included in our analysis models, the residual confounding effects due to the unmeasured or incompletely measured factors cannot be excluded.

## 5. Conclusions

In summary, an appropriate plasma retinol concentration and a greater intake of dietary VA and β-carotene may result in better bone health status among children who are aged 6–9 years, according to our study.

## Figures and Tables

**Figure 1 nutrients-14-04694-f001:**
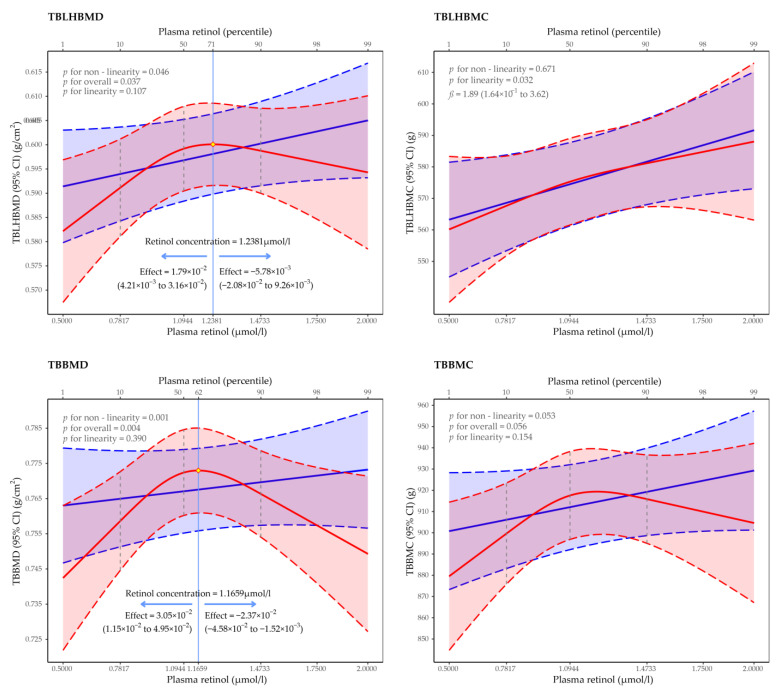
Association of predicted plasma retinol concentration with bone mineral indexes in children who were aged 6–9 years. Average predicted values of bone mineral indexes are indicated by solid lines, and 95% CI are indicated by shaded areas with lower, and upper values are indicated by dash lines (red: non-linearity, blue: linearity). Knots of plasma retinol concentration were placed at 10th (0.7817 μmol/L), 50th (1.0944 μmol/L) and 90th (1.4733 μmol/L) centiles according to the distribution of data, and these are indicated by vertical dash segment. The predicted maximum values of TBLH BMD and TB BMD are indicated as yellow diamond and blue vertical line, respectively. TB, total body; TBLH, total body less head; BMD, bone mineral density; BMC, bone mineral content.

**Table 1 nutrients-14-04694-t001:** Characteristics of 426 participants divided by knots of 10th, 50th, 70th and 90th centiles of plasma retinol concentration.

Variables	Group 1 (*n* = 42)	Group 2 (*n* = 171)	Group 3 (*n* = 85)	Group 4 (*n* = 86)	Group 5 (*n* = 42)	*p*	Total (*n* = 426)
Age (years)	7.93 (0.80)	7.94 (1.00)	8.07 (1.00)	8.19 (0.96)	8.17 (1.10)	0.291	8.0 (1.0)
Gender (*n*, %)						0.514	
Girls	17 (40.5)	68 (39.8)	37 (43.5)	38 (44.2)	23 (54.8)		183 (43.0)
Boys	25 (59.5)	103 (60.2)	48 (56.5)	48 (55.8)	19 (45.2)		243 (57.0)
Height (cm)	127.3 (7.7)	127.9 (7.9)	128.2 (7.4)	130.4 (7.8)	129.6 (10.6)	0.112	128.5 (8.1)
Weight (kg)	24.4 (4.5)	25.7 (7.1)	25.9 (5.9)	27.8 (6.9)	28.3 (9.8)	**0.022**	26.3 (7.0)
BMI (kg/m^2^)	15.0 (1.4)	15.6 (2.7)	15.6 (2.4)	16.2 (2.7)	16.6 (3.4)	**0.021**	15.7 (2.7)
MET [kcal/(kg·day)]	39.9 (4.5)	39.8 (4.3)	40.1 (4.5)	40.1 (4.5)	40.0 (4.4)	0.988	40.0 (4.4)
Energy (kcal/day)	1333 (418)	1404 (440)	1404 (377)	1545 (452)	1455 (445)	0.055	1430 (432)
Protein (g/day)	59 (22)	62 (21)	64 (23)	72 (26)	67 (22)	**0.009**	65 (23)
Fat (g/day)	38 (14)	43 (21)	44 (21)	47 (19)	44 (17)	0.255	44 (19)
Carbohydrate (g/day)	193 (63)	196 (58)	191 (40)	214 (63)	203 (60)	0.062	199 (57)
Calcium (mg/day)	456 (182)	486 (208)	492 (209)	583 (219)	529 (220)	**0.003**	508 (212)
Phosphorus (mg/day)	917 (321)	970 (301)	986 (303)	1119 (355)	1034 (312)	**0.002**	1004 (321)
Delivery mode (*n*, %)						0.762	
Vaginal	20 (47.6)	85 (49.7)	47 (55.3)	39 (45.3)	20 (47.6)		211 (49.5)
Cesarean	22 (52.4)	86 (50.3)	38 (44.7)	47 (54.7)	22 (52.4)		215 (50.5)
Feeding patterns (*n*, %)						0.457	
Breastfeeding	34 (81.0)	146 (85.4)	78 (91.8)	72 (83.7)	36 (85.7)		366 (85.9)
Artificial feeding	8 (19.0)	25 (14.6)	7 (8.2)	14 (16.3)	6 (14.3)		60 (14.1)
Household income (*n*, %)						0.346	
<8000 Yuan/month	8 (19.0)	35 (20.5)	13 (15.3)	15 (17.4)	5 (11.9)		76 (17.8)
8000~15,000 Yuan/month	13 (31.0)	55 (32.2)	25 (29.4)	24 (27.9)	13 (31.0)		130 (30.5)
>15,000 Yuan/month	12 (28.6)	50 (29.2)	26 (30.6)	36 (41.9)	20 (47.6)		144 (33.8)
Unknown	9 (21.4)	31 (18.1)	21 (24.7)	11 (12.8)	4 (9.5)		76 (17.8)
Maternal education level (*n*, %)						0.149	
Secondary or less	20 (47.6)	76 (44.4)	29 (34.1)	23 (26.7)	13 (31.0)		161 (37.8)
University	18 (42.9)	84 (49.1)	49 (57.6)	53 (61.6)	26 (61.9)		230 (54.0)
Postgraduate or above	4 (9.5)	11 (6.4)	7 (8.2)	10 (11.6)	3 (7.1)		35 (8.2)
Paternal education level						0.246	
Secondary or less	24 (57.1)	79 (46.2)	31 (36.5)	28 (32.6)	15 (35.7)		177 (41.5)
University	14 (33.3)	72 (42.1)	42 (49.4)	46 (53.5)	22 (52.4)		196 (46.0)
Postgraduate or above	4 (9.5)	20 (11.7)	12 (14.1)	12 (14.0)	5 (11.9)		53 (12.4)
Use of calcium supplement (*n*, %)						0.900	
No	25 (59.5)	102 (59.6)	56 (65.9)	53 (61.6)	25 (59.5)		261 (61.3)
Yes	17 (40.5)	69 (40.4)	29 (34.1)	33 (38.4)	17 (40.5)		165 (38.7)
Use of multivitamin supplement (*n*, %)						0.521	
No	38 (90.5)	139 (81.3)	67 (78.8)	72 (83.7)	36 (85.7)		352 (82.6)
Yes	4 (9.5)	32 (18.7)	18 (21.2)	14 (16.3)	6 (14.3)		74 (17.4)
BMC (g)							
Total body	882 (105)	919 (128)	927 (129)	967 (149) ^a^	951 (178)	**0.009**	930 (138)
Total body less head	554 (87)	572 (103)	581 (95)	616 (120) ^a,b^	615 (154)	**0.004**	585 (111)
BMD (g/cm^2^)							
Total body	0.757 (0.047)	0.778 (0.060)	0.784 (0.063)	0.797 (0.063) ^a^	0.783 (0.077)	**0.014**	0.781 (0.063)
Total body less head	0.590 (0.050)	0.602 (0.061)	0.610 (0.058)	0.627 (0.062) ^a,b^	0.621 (0.088)	**0.006**	0.609 (0.063)
TBF (kg)	6.3 (2.0)	7.3 (3.6)	7.2 (3.1)	8.0 (3.9)	8.7 (4.7)	**0.015**	7.4 (3.6)
Plasma retinol concentration (μmol/L)	0.67 (0.11)	0.95 (0.09)	1.16 (0.04)	1.34 (0.08)	1.65 (0.19)	**<0.001**	1.11 (0.28)
Vitamin A (μg RE/day)	550 (280)	648 (344)	642 (287)	768 (424)	795 (366)	**0.001**	676 (354)
α-carotene (μg /day)	763 (882)	823 (879)	814 (692)	1004 (1086)	905 (788)	0.491	860 (883)
β-carotene (μg /day)	2038 (1401)	2343 (1648)	2234 (1320)	2817 (1958)	2559 (1301)	0.052	2408 (1615)
β-cryptoxanthin (μg/day)	86.5 (75.5)	91.4 (84.8)	93.6 (76.0)	110.3 (113.5)	116.4 (186.8)	0.417	97.6 (103.0)

Group 1, children whose plasma retinol concentration value was at first 10th; Group 2, children whose plasma retinol concentration value was between 10th and 50th; Group 3, children whose plasma retinol concentration value was between 50th and 70th; Group 4, children whose plasma retinol concentration value was between 70th and 90th; Group 5, children whose plasma retinol concentration value was at last 10th; BMI, body mass index; MET, metabolic equivalent; BMD, bone mineral density; BMC, bone mineral content; TBF, total body fat; RE, retinol equivalent. Bonferroni method was used for pairwise comparison, ^a^ compared with group 1, *p* < 0.05; ^b^ compared with group 2, *p* < 0.05.

**Table 2 nutrients-14-04694-t002:** Multiple linear regression analysis between dietary VA intake and bone mineral status.

Variables	Per SD Increase in Dietary VA Intake of Different Forms					
	Model 1			Model 2		
	*β*	95% CI	*p*	*β*	95% CI	*p*
**Dietary VA intake**						
TB BMC (g)	13.2	(2.75, 23.6)	0.001	6.16	(−1.11, 13.4)	0.098
TB BMD (g/cm^2^)	9.21 × 10^−3^	(4.21 × 10^−3^, 1.42 × 10^−2^)	<0.001	**5.86 × 10^−3^**	**(1.56 × 10^−3^, 1.02 × 10^−2^)**	**0.008**
TBLH BMC (g)	7.119	(−1.27, 15.5)	0.097	2.53	(−2.31, 7.37)	0.306
TBLH BMD (g/cm^2^)	5.79 × 10^−3^	(8.13 × 10^−4^, 1.07 × 10^−2^)	0.022	3.04 × 10^−3^	(−3.72 × 10^−5^, 6.12 × 10^−3^)	0.053
**Dietary α-carotene intake**						
TB BMC (g)	18.9	(−5.60, 43.4)	0.131	13.2	(−3.03, 29.4)	0.111
TB BMD (g/cm^2^)	1.22 × 10^−2^	(4.20 × 10^−3^, 2.40 × 10^−2^)	0.043	9.10 × 10^−3^	(−5.04 × 10^−4^, 1.87 × 10^−2^)	0.064
TBLH BMC (g)	12.8	(−6.78, 32.4)	0.201	9.73	(−1.03, 20.5)	0.077
TBLH BMD (g/cm^2^)	8.96 × 10^−3^	(−2.64 × 10^−3^, 2.06 × 10^−2^)	0.131	6.80 × 10^−3^	(−4.04 × 10^−5^, 1.36 × 10^−2^)	0.052
**Dietary β-carotene intake**						
TB BMC (g)	17.3	(6.36, 28.2)	0.002	5.97	(−1.48, 13.42)	0.117
TB BMD (g/cm^2^)	9.16 × 10^−3^	(3.91 × 10^−3^, 1.41 × 10^−2^)	<0.001	**4.47 × 10^−3^**	**(5.47 × 10^−5^, 8.88 × 10^−2^)**	**0.048**
TBLH BMC (g)	13.1	(4.36, 21.8)	0.003	3.92	(−1.02, 8.86)	0.121
TBLH BMD (g/cm^2^)	8.42 × 10^−3^	(3.25 × 10^−3^, 1.36 × 10^−2^)	0.001	3.03 × 10^−3^	(−1.06 × 10^−4^, 6.17 × 10^−3^)	0.059
**Dietary β-cryptoxanthin intake**						
TB BMC (g)	−5.35	(−15.9, 5.19)	0.321	−4.29	(−11.3, 2.67)	0.228
TB BMD (g/cm^2^)	−1.85 × 10^−3^	(−6.95 × 10^−3^, 3.25 × 10^−3^)	0.476	−1.27 × 10^−3^	(−5.41 × 10^−3^, 2.87 × 10^−3^)	0.549
TBLH BMC (g)	−5.20	(−13.63, 3.23)	0.227	−4.28	(−8.89, 0.33)	0.070
TBLH BMD (g/cm^2^)	−2.79 ×10^−3^	(−7.79 × 10^−3^, 2.21 × 10^−3^)	0.274	−2.13 × 10^−3^	(−5.07 × 10^−3^, 8.10 × 10^−4^)	0.157

Model 1: adjusted for age and sex; Model 2: adjusted for height, TBF, delivery mode, household income, feeding patterns, parental educational levels, physical activity, use of calcium and multivitamin supplements, energy adjusted dietary intakes of total energy, protein, calcium and phosphorus based on Model 1.

## Data Availability

Not applicable.

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
