# Peer review of "Vitamin A Nutritional Status Is a Key Determinant of Bone Mass in Children"

_nutrients, 2022, doi:10.3390/nu14214694_

Round 1

Reviewer 1 Report

Thank you for the opportunity to review this intersting manuscript.

Vitamin A is vital to child health and immune function; hence, in settings where vitamin A deficiency is a public health problem, vitamin A supplementation is recommended in infants and children aged 6-59 months as a public health intervention to reduce child morbidity and mortality. The aim of this study was to assess whether plasma retinol concentration,  dietary VA intake, and carotenoids intake are positively associated with bone mineral sta-57 tus among healthy children aged 6–9 years.

I have some suggestion to improve this manuscript. The introduction could be implemented. In particular I think that it's necessary to need to emphasize more to the correlation between VA intake and  bone mineral status among healthy children with other studies that analyzed this problem. In addition it might be useful mentioned other possible benefits form VA suplementations.  It is important to vision, growth, cell division, reproduction and immunity.Here some references "Sinopoli, A.; Caminada, S.;Isonne, C.; Santoro, M.M.; Baccolini,V. What Are the Effects of Vitamin A
Oral Supplementation in the Prevention and Management of Viral
Infections? A Systematic Review of Randomized Clinical Trials. Nutrients 2022, 14, 4081. https://doi.org/10.3390/nu14194081";"Villamor, E.; Fawzi, W.W. Effects of Vitamin A Supplementation on Immune Responses and Correlation with Clinical Outcomes.Clin. Microbiol. Rev. 2005, 18, 446–464".

Methods are well structured.

In the conclusion I suggest to add "in our study".

In general I think it's a good study. I ask the authors to add my suggestions to make it more complete, please.

Author Response

Response to Reviewer 1 Comments

Manuscript ID: nutrients-1972635

Title: Vitamin A Nutritional Status is a Key Determinant of Bone Mass in Children

Dear Editor:

We have revised our manuscript mainly according to your comments and suggestions, and answered the reviewer’s all questions point by point. Please kindly refer to the revised manuscript and see Response to Reviewers enclosed below for the details. We greatly appreciate the reviewer’s thoughtful advice and comments to help improve our study. Thank you for your attention.

Sincerely,

Zheqing Zhang & Yongnong Ye

Point 1: Vitamin A is vital to child health and immune function; hence, in settings where vitamin A deficiency is a public health problem, vitamin A supplementation is recommended in infants and children aged 6-59 months as a public health intervention to reduce child morbidity and mortality. The aim of this study was to assess whether plasma retinol concentration, dietary VA intake, and carotenoids intake are positively associated with bone mineral status among healthy children aged 6–9 years. I have some suggestion to improve this manuscript. The introduction could be implemented. In particular I think that it's necessary to need to emphasize more to the correlation between VA intake and bone mineral status among healthy children with other studies that analyzed this problem. In addition it might be useful mentioned other possible benefits form VA supplementations.  It is important to vision, growth, cell division, reproduction and immunity. Here some references:1. "Sinopoli, A.; Caminada, S.; Isonne, C.; Santoro, M.M.; Baccolini,V. What Are the Effects of Vitamin A Oral Supplementation in the Prevention and Management of Viral Infections? A Systematic Review of Randomized Clinical Trials. Nutrients 2022, 14, 4081. https://doi.org/10.3390/nu14194081"; 2. "Villamor, E.; Fawzi, W.W. Effects of Vitamin A Supplementation on Immune Responses and Correlation with Clinical Outcomes.Clin. Microbiol. Rev. 2005, 18, 446–464".

Response 1: Thanks for your valuable suggestions. We systemically retrieved the articles again for the relationship between bone mineral status and VA in children but the results showed that there was no related articles. Most of the researches published up to now were conducted among adults. This is the main reason that we performed this study. As you suggested, we mentioned other benefits of VA on health in the section of Introduction as below:

“VA, mainly derived from animal sources such as liver, dairy, and egg yolks or plant-based sources enriched with carotenoids such as carrot, mango, and pumpkin (e.g., α-carotene, β-carotene, and β-cryptoxanthin), is a fat-soluble antioxidant and an essential nutrient which is of great importance on improving children’s innate and adaptive immune response (lower susceptibility to infections, reduction of complications, and better prognosis), maintaining normal vision, enhancement of growth, cell differentiation and reproduction[1-6].”

Point 2: Methods are well structured.

Response 2: Thank you for your appreciation.

Point 3: In the conclusion I suggest to add "in our study".

Response 3: Added, thank you.

Point 4: In general I think it's a good study. I ask the authors to add my suggestions to make it more complete, please.

Response 4: We have revised the manuscript according to your valuable suggestions, thank you.

References

  1. Fairfield, K.M.; R.H. Fletcher. Vitamins for chronic disease prevention in adults: scientific review. Jama 2002, 287(23), p. 3116-3126.
  2. Gerster, H. Vitamin A--functions, dietary requirements and safety in humans. Int J Vitam Nutr Res 1997, 67(2), p. 71-90.
  3. Sinopoli, A.; S. Caminada; C. Isonne; M.M. Santoro; V. Baccolini. What Are the Effects of Vitamin A Oral Supplementation in the Prevention and Management of Viral Infections? A Systematic Review of Randomized Clinical Trials. Nutrients 2022, 14(19).
  4. Stephensen, C.B. Vitamin A, infection, and immune function. Annu Rev Nutr 2001, 21, p. 167-192.
  5. Tanumihardjo; A. Sherry. Carotenoids and Bone Health. Humana Press 2013, 10.1007/978-1-62703-203-2(Chapter 14), p. 237-245.
  6. Villamor, E.; W.W. Fawzi. Effects of vitamin a supplementation on immune responses and correlation with clinical outcomes. Clin Microbiol Rev 2005, 18(3), p. 446-464.

Reviewer 2 Report

The investigators studied the relationship between vitamin A and BMD/BMC in children and found an inverse U shaped relationship between retinol and BMD.

1. There is no mention of institutional review board approval and details of how the consent was obtained from the subjects and parents.

2. Although, the statistical analyses which were performed suggest an inverse U shaped relationship for some of the parameters, the BMD/BMC in the top 10th percentiles are not lower than the other groups (Table 1). A simple plot showing the relationship between the BMD and plasma retinol levels would be helpful. A fuller explanation of the inverse U relationship is needed

3. Table 1 Heading- Tertiles is not the correct description of the data presentation

4. Figure 1- The correct label is percentiles (not percent) 

Author Response

Response to Reviewer 2 Comments

Manuscript ID: nutrients-1972635

Title: Vitamin A Nutritional Status is a Key Determinant of Bone Mass in Children

Dear Editor:

We have revised our manuscript mainly according to your comments and suggestions, and answered the reviewer’s all questions point by point. Please kindly refer to the revised manuscript and see Response to Reviewers enclosed below for the details. We greatly appreciate the reviewer’s thoughtful advice and comments to help improve our study. Thank you for your attention.

Sincerely,

Zheqing Zhang & Yongnong Ye

Point 1: The investigators studied the relationship between vitamin A and BMD/BMC in children and found an inverse U-shaped relationship between retinol and BMD. There is no mention of institutional review board approval and details of how the consent was obtained from the subjects and parents.

Response 1: Thank you for your comments. We have added relevant details in the Subjects part of Materials and Methods section as follows:

“This cross-sectional study was approved by the ethics committee of the School of Public Health at Sun Yat-sen University (No. 201549). A well-written consent was obtained before enrollment for each participant through his or her parent or legal guardian.”

Point 2: Although, the statistical analyses which were performed suggest an inverse U-shaped relationship for some of the parameters, the BMD/BMC in the top 10th percentiles are not lower than the other groups (Table 1). A simple plot showing the relationship between the BMD and plasma retinol levels would be helpful. A fuller explanation of the inverse U relationship is needed.

Response 2: Based on the linear and non-linear analysis of the data, we detected an inverse U-shaped relationship for the retinol level in blood and TB BMD/TBLH BMD. And the highest values of TB BMD/TBLH BMD was obtained in children when the retinol level was equal to around 70th percentile. Thus, TB BMD/TBLH BMD in the top 10th percentiles was lower than the 70th percentile but not other groups. In the revised Table 1, we presented the mean (SD) values for the variables of 50th-70th and 70th-90th percentile and applied Bonferroni method for pairwise comparison. VA is a fat-soluble vitamin which may exert negative impacts on our body including bone health when we consume more than the amount we need. Therefore, we applied both linear and non-linear analysis to comprehensively investigate the association of retinol levels and bone mineral status in children as showed in Figure 1. Following your suggestion, we revised the explanation for inverse U relationship in the Mechanisms part of the Discussion section as below:

“The following mechanisms may explain the associations observed in our study. Adequate VA (retinol or retinoic acid) is of great importance in maintaining bone homeostasis, which effectively suppresses the over-excretion of parathyroid hormone[1, 2]. It may also induce the secretion of growth hormone to stimulate the production of insulin-like growth factor 1; both these compounds promote osteoblast proliferation and differentiation and reduce osteoblast apoptosis by stabilizing β-catenin in the Wnt signaling pathway[3-5]. However, VA is fat-soluble vitamin, long-term excessive intake will accumulate in the body and cause toxic side effects such as alopecia, cheilosis, liver injury and so on[6, 7]. Previous studies in vivo and in vitro have indicated that high concentration of preformed VA negatively affects both osteoblast differentiation and mineralization owing to the effects of VA on osteogenic gene inhibition, osteoclastogenic gene activation, and osteocyte or osteoblast-related peptides modulation[8]. Through an animal experiment using female C57BL6/J mouse models, researchers reported a significant increase in osteoclast number and osteoclastogenic gene expression in mice treated with high doses of dietary retinyl acetate[9]. Moreover, VA antagonizes vitamin D (VD) in increasing calcium absorption and maintaining steady-state serum calcium concentration. Retinoic acid and 1, -25-hydroxyvitamin D share a common nuclear receptor that interacts with both the retinoic acid receptor and VD receptor. Therefore, high plasma retinol concentration may reduce the function of VD and negatively affect bone homeostasis[10]. These facts may provide potential explanations for the inversed U relationship between plasma retinol concentration and bone mineral status in our study. A majority of previous studies suggested that oxidative stress caused by reactive oxygen species can promote bone resorption and inhibit osteogenic differentiation[11]. These processes can negatively affect bone homeostasis, reduce bone mass, and even lead to osteoporosis[12]. Carotenoids act as antioxidants that quench singlet oxygen and trap peroxyl radicals, reducing oxidative stress and potentially reducing bone resorption, thereby protecting the bone tissue[13, 11]. However, the oxidation capacities of various carotenoids are different[14]. β-carotene is one of the most abundant carotenoids in the diet who has the highest biological activities for conversion to retinol equivalents[15]. It can also inhibit osteoclastogenesis and bone resorption by suppressing the receptor activator of nuclear factor kappa-B ligand signaling pathway[16]. These mechanisms may collectively contribute to the protective effects of dietary carotenoids intake (especially for β-carotene) on bone health.”

Point 3: Table 1 Heading- Tertiles is not the correct description of the data presentation.

Response 3: Revised, thank you.

Point 4: Figure 1- The correct label is percentiles (not percent).

Response 4: Revised, thank you.

References

  1. Paik, J.M.; W.R. Farwell; E.N. Taylor. Demographic, dietary, and serum factors and parathyroid hormone in the National Health and Nutrition Examination Survey. Osteoporos Int 2012, 23(6), p. 1727-1736.
  2. Pluijm, S.M.; M. Visser; J.H. Smit; C. Popp-Snijders; J.C. Roos; P. Lips. Determinants of bone mineral density in older men and women: body composition as mediator. J Bone Miner Res 2001, 16(11), p. 2142-2151.
  3. Djakoure, C.; J. Guibourdenche; D. Porquet; P. Pagesy; F. Peillon; J.Y. Li; D. Evain-Brion. Vitamin A and retinoic acid stimulate within minutes cAMP release and growth hormone secretion in human pituitary cells. J Clin Endocrinol Metab 1996, 81(8), p. 3123-3126.
  4. Locatelli, V.; V.E. Bianchi. Effect of GH/IGF-1 on Bone Metabolism and Osteoporsosis. Int J Endocrinol 2014, 2014, p. 235060.
  5. Raifen, R.; Y. Altman; Z. Zadik. Vitamin A levels and growth hormone axis. Horm Res 1996, 46(6), p. 279-281.
  6. Dawson, M.I. The importance of vitamin A in nutrition. Curr Pharm Des 2000, 6(3), p. 311-325.
  7. Russell, R.M. The vitamin A spectrum: from deficiency to toxicity. Am J Clin Nutr 2000, 71(4), p. 878-884.
  8. Yee, M.M.F.; K.Y. Chin; S. Ima-Nirwana; S.K. Wong. Vitamin A and Bone Health: A Review on Current Evidence. Molecules 2021, 26(6).
  9. Lionikaite, V.; P. Henning; C. Drevinge; F.A. Shah; A. Palmquist; P. Wikström; S.H. Windahl; U.H. Lerner. Vitamin A decreases the anabolic bone response to mechanical loading by suppressing bone formation. Faseb j 2019, 33(4), p. 5237-5247.
  10. Johansson, S.; H. Melhus. Vitamin A antagonizes calcium response to vitamin D in man. J Bone Miner Res 2001, 16(10), p. 1899-1905.
  11. Tanumihardjo; A. Sherry. Carotenoids and Bone Health. Humana Press 2013, 10.1007/978-1-62703-203-2(Chapter 14), p. 237-245.
  12. Sánchez-Rodríguez, M.A.; M. Ruiz-Ramos; E. Correa-Muñoz; V.M. Mendoza-Núñez. Oxidative stress as a risk factor for osteoporosis in elderly Mexicans as characterized by antioxidant enzymes. BMC Musculoskelet Disord 2007, 8, p. 124.
  13. Stahl, W.; H. Sies. Antioxidant activity of carotenoids. Mol Aspects Med 2003, 24(6), p. 345-351.
  14. El-Agamey, A.; G.M. Lowe; D.J. McGarvey; A. Mortensen; D.M. Phillip; T.G. Truscott; A.J. Young. Carotenoid radical chemistry and antioxidant/pro-oxidant properties. Arch Biochem Biophys 2004, 430(1), p. 37-48.
  15. Saini, R.K.; S.H. Nile; S.W. Park. Carotenoids from fruits and vegetables: Chemistry, analysis, occurrence, bioavailability and biological activities. Food Res Int 2015, 76(Pt 3), p. 735-750.
  16. Wang, F.; N. Wang; Y. Gao; Z. Zhou; W. Liu; C. Pan; P. Yin; X. Yu; M. Tang. β-Carotene suppresses osteoclastogenesis and bone resorption by suppressing NF-κB signaling pathway. Life Sci 2017, 174, p. 15-20.

Round 2

Reviewer 2 Report

minor language review and corrections needed e.g., line 228; over secretion not 'excretion'.